# The Executive Skills Questionnaire-Revised: Adaptation and Psychometric Properties in the Working Context of Malaysia

**DOI:** 10.3390/ijerph18178978

**Published:** 2021-08-26

**Authors:** Hira Nasir, Chee-Seng Tan, Kai-Shuen Pheh

**Affiliations:** Department of Psychology and Counselling, Universiti Tunku Abdul Rahman (UTAR), Kampar 31900, Perak, D. R., Malaysia; hira.nasir@1utar.my (H.N.); phehks@utar.edu.my (K.-S.P.)

**Keywords:** executive functioning, cognitive processes, confirmatory factor analysis, reliability, psychometric

## Abstract

Executive functions (EFs) are a set of high-level cognitive and behavioral monitoring skills that are important to employees’ work performance. The 25-item Executive Skills Questionnaire-Revised (ESQ-R) measures executive dysfunction in five dimensions (e.g., emotional regulation). Nevertheless, the usability of this newly developed scale for employees remains unclear. The present study evaluated the psychometric properties of the adopted ESQ-R for working adults in Malaysia. A total of 325 employees responded to an online survey consisted of the ESQ-R, Executive Function Index (EFI), self-rated creativity scale (SRCS), and 9-item Utretch Work Engagement Scale (UWES-9) and Employee Well-being Scale. Several CFAs were conducted to compare three competing models. While all models showed a good fit, the 5-factor second-order model that is in line with the theoretical structure is preferable. The ESQ-R showed excellent internal consistency. Moreover, the ESQ-R score was negatively correlated with EFI, creativity, and UWES-9 scores, supporting the convergent, discriminant, and concurrent validity. The ESQ-R score also explained incremental variance in well-being above and beyond scores of the UWES-9 and SRCS. Taken together, the ESQ-R is a useful tool for assessing employees’ executive dysfunction and suggesting intervention programs helping employees with deficits in EFs.

## 1. Introduction

Executive functions (EFs) refer to intellectual behaviors such as planning, self-monitoring, problem-solving, reasoning, and working memory that stem from the dorsolateral prefrontal cortex of the brain [1,2]. EFs are a control mechanism that work in regulating human cognition and behaviors [2] and are necessary when an individual wants to exercise higher cognitive skills. Studies have shown that EFs are positively associated with well-being (e.g., [3,4]), mindfulness [5], and resilience [6], and negatively associated mental health (e.g., [7]), just to name a few.

In addition, EFs are a core component of self-control or self-regulation [2], which is important to working performance [8]. Moreover, EFs play a critical role in the development of other essential competencies including economic decision making, risk-taking behaviors, the judgment of another’s intention, and level of social trust [9,10,11] as well as self-management of time and self-regulation of emotions [12]. Therefore, it is reasonable to believe that EFs are conducive to employees’ performance [9]. Indeed, EFs have been found beneficial to working performance and work engagement [13,14,15,16,17].

Although EFs play a critical role in employees’ performance, little attention has been given to the psychometric qualities of the measurement of EFs in the organizational context. This study, therefore, aims to address this gap by investigating the psychometric properties of the recently developed Executive Skills Questionnaire-Revised (ESQ-R) [18] in a sample of working adults in Malaysia. The findings are expected to shed light on the usefulness of the ESQ-R in measuring EFs for working adults. In the following sections, we first reviewed the concept of EFs and their influence on work performance and followed by a brief review of the measurement of EFs.

### 1.1. The Role of Executive Functions in Working Contexts

Historically, EF is commonly studied in the clinical neuropsychology context, particularly among people with traumatic brain injury, dementia, and other cognitive disorders [19,20,21]. Research on EFs is a burgeoning field, hence literature in the healthy adult population, let alone the working population, is relatively scarce. In the workplace, EFs refer to the interaction between different cognitive and emotional abilities (e.g., perceptions, and reasoning) to perform daily functions such as planning, tasks shifting, and problem solving [22]. 

EFs such as planning, attention, and cognitive flexibility are important to job success [23,24]. For example, time management planning improves employees’ performance because employees with good time management are good at prioritizing their tasks [16]. Meanwhile, effective regulation of attention helps employees to focus on the tasks at hand, hence avoiding inappropriate mind wandering, therefore improving work performance [25,26]. Furthermore, Castellano et al. [14] found that employees who use elaborate processes of emotional regulation (e.g., taking criticism in a positive light) tend to have greater work engagement and less burnout.

Besides, EFs enable employees to regulate their emotions and interact effectively in complex environments [9,25]. According to Chan et al. [25], EFs matter more to employees who are sparse on time and are required to make unexpected but effective decisions on the spot. When in a time crunch, or a novel situation, EFs allow individuals to direct their attention to the problems and think flexibly to generate a solution. Therefore, it is believed that EFs are critical to executives who are often required to switch between different roles, make spontaneous decisions, and solve problems in uncertain and diverse contexts, as well as communicate effectively with co-workers. Individuals with deficits in executive functioning are likely to encounter difficulties in completing tasks that require mental control and have poor productivity and hence, they have a lower chance of finding and keeping an executive-level job [23,24]. 

### 1.2. Executive Functions and Creativity

In addition to work performance, studies (e.g., [27,28]) have found an association between the different components of EFs and creativity—the generation of new and useful ideas, solutions, and products [29,30,31,32]. Individuals who displayed greater cognitive flexibility tasks were able to produce more new responses [33]. Zabelina and colleagues [28] have found that artists who have greater overall EFs and cognitive flexibility can regulate their thoughts and behaviors as well as shifting between goals and ideas, respectively. Similarly, Benedek et al. [34] found that cognitive inhibition correlates positively with various measures of creativity including indicators of divergent thinking. This could be due to cognitive inhibition promoting the generation of new ideas by disregarding irrelevant responses [34]. 

### 1.3. Executive Functions and Work Engagement

The different components of EFs have also been found to have a positive relationship with work engagement, the “positive, fulfilling, work-related state of mind” [35] (p. 74). For instance, Parke et al. [16] found that time management (a factor of EFs) improved daily performance at work by enhancing work engagement. Moreover, it was found that employees demonstrate greater engagement at work when they use elaborate processes of emotional regulation [14].

Taken together, organizations and researchers are suggested to use executive functioning abilities assessments for employee selection and promotion [25], as individuals with better EFs have a better mastery of workplace tasks. Accurately assessing EFs of employees also helps in recognizing talent and ensuring employees’ retainment within the organization, therefore it is encouraged to investigate and develop measures of EF that are applicable and practical in the organizational setting [25].

### 1.4. Measurement of EFs

Several measures have been developed to assess EFs. For instance, the Behaviour Rating Inventory of Executive Function-Adult Version [36] and Barkley Deficits in Executive Functioning Scale [37]. Note that, however, those measures are usually catered toward the clinical population and are costly. 

Apart from those mentioned above, there are measurement tools of EFs that cater toward the nonclinical population. The Adult Executive Functioning Inventory (ADEXI) [38] and the Executive Function Index (EFI) [39] are two of those. While ADEXI exhibits sufficient psychometric properties, it is unfortunately technically inadequate as a sole measurement tool of EF. This is because the ADEXI requires multiple ratings and is used as a complement to neuropsychological tools [38]. The EFI, on the other hand, is a self-rated executive functioning measure with 27 items and good internal consistency [39]. Although the EFI has been validated in different settings, the scale is not built as an intervention-focused measure.

Strait et al. [18] report that the available adult EF rating scales are either technically inadequate (but highly efficient and accessible) or have strong technical adequacy but require extensive training to administer it and are costly. Most of the EF measurements are intended and validated for the clinical population with neurodevelopmental or neurodegenerative disorders. These measurements are typically too pathological-oriented to be used with the nonclinical population [39]. To overcome the limitation, Dawson and Guare [40,41,42] developed four versions of the Executive Skills Questionnaire (ESQ) to provide an efficient and affordable self-assessment of EF for children (33 items), teenagers (33 items), students (33 items), and adults (36 items). All the versions of ESQ are designed to assess the 11 areas of EF skills such as goal-directed persistence, response inhibition, working memory, and sustained attention. 

Strait et al. [18] refined the ESQ and developed the Executive Skills Questionnaire-Revised (ESQ-R) for the young adult population and concerning academic success. The ESQ-R consists of 25 items that focus on five dimensions namely plan management, time management, material organization, emotional regulation, and behavioral regulation. The ESQ-R has two advantages. First, the multidimensional ESQ-R has fewer items than other EFs measures. Hence, the ESQ-R requires less cognitive and physical burden from participants but offering a comprehensive insight into their EFs. In addition, similar to the ESQ, the ESQ-R is directly tied to the available intervention guides for EFs. In other words, the results derived from ESQ-R can be used to design appropriate intervention plans according to the needs of the participants.

Although the ESQ-R is initially designed to focus on EF dimensions that apply to young adults in the academic contexts (e.g., time management), it is reasonable to believe that the ESQ-R is also applicable for working adults in the working contexts. This is because, when facing new and uncertain task demands, employees not only need pre-existing knowledge but also a different set of skills such as plan management and material organization (foreseeing future challenges that might be faced by or within the organization and having contingencies in place), emotional and behavioral regulation (managing partners and employees, juggling multiple roles, and dealing with uncertain and stressful situations), and time management (coming up with a solution or decision making on the spot) to achieve their goals [25]. 

### 1.5. The Present Study

Although EF plays a critical role in employees’ performance and the development of organizations, assessment of EF in the organizational context has long been neglected. This might be due to the misunderstanding that EF and general intelligence are interchangeable, and hence assessing EF within organizations is impractical [25]. The recently developed Executive Skills Questionnaire-Revised (ESQ-R) by Strait et al. [18] measures deficits of executive skills among adults. It is believed that the ESQ-R could be a potential measurement of employees’ EF. Nevertheless, to date, no study has been conducted on the psychometric qualities of the newly developed ESQ-R in the organizational context. Hence, it is unclear whether the ESQ-R is an appropriate measurement of EF for healthy working adults. This study, therefore, aims to address this gap by investigating the psychometric properties of the ESQ-R in a sample of working adults in Malaysia by examining its factorial structure and internal consistency. We also examined the construct and criterion validity of the ESQ-R. Construct validity was tested by correlating the ESQ-R score with the Executive Function Index (EFI) [39] score for convergent validity and self-rated creativity for discriminant validity. Both concurrent and incremental validity was used to shed light on the criterion validity. The ESQ-R score was correlated with work engagement for testing concurrent validity. Meanwhile, incremental validity was investigated by testing the extent to which ESQ-R score can explain employee well-being beyond work engagement and creativity. The rationales for choosing these variables are presented in the Analytical Plan. 

## 2. Method

### 2.1. Research Design and Participants

The research was conducted using a quantitative approach and cross-sectional design. Homogenous purposive sampling was used to recruit participants using online surveys. The inclusion criteria were full-time executive professionals working in Malaysia with a minimum of one year of work experience. According to the Public Services Commission of Malaysia [43], any professional working under the job category Grade 41 and above is considered an executive (Management and Professional Group). Examples of executive professionals are lecturers, engineers, journalists, and landscape architects. 

After obtaining the ethical approval from the Scientific and Ethical Review Committee of Universiti Tunku Abdul Rahman (ref: U/SERC/192/2019), an online survey developed using Qualtrics was distributed through email and social networking sites such as Facebook, LinkedIn, and WhatsApp. A total of 458 responses were collected. However, 133 responses were excluded due to not meeting the criteria (e.g., non-executive, part-timer, not currently working in Malaysia), not answering the survey at all, or not agreeing to participate in the study. The total number of responses retained for analysis was 325 (153 males and 171 females, 1 missing value). The age range of the participants was 23 to 80 years old (M = 40.34, SD = 10.39). The sample consisted of 37.2% Malays, 32.3% Chinese, 14.8% Indians, and 15.7% Others. Of the sample, 275 were Malaysian while 50 were foreigners currently working in Malaysia. The majority (70.8%) of the participants worked in the academic setting, 8.6% of the participants were from business and administration, 8% worked in management positions while the remaining were from other fields (healthcare, hospitality, legal, science and engineering, and social and cultural). All participants gave their consent before answering the survey. 

### 2.2. Measurements

Executive Skills Questionnaire-Revised (ESQ-R) [18]. The ESQ-R consists of 25 items to measure executive deficiency on a 4-point frequency-based response scale from never or rarely (0) to very often (3). The ESQ-R has five factors: plan management, time management, materials organization, emotional regulation, and behavioral regulation. Plan management (11 items) refers to individuals’ ability to accomplish a task by creating and managing a plan. A sample item is “I have trouble with tasks where I have to come up with my own ideas”. Time management (4 items) is the skill area that refers to the ability to manage various aspects of time such as time approximation, time allotment, and working within time limits or constraints. The sample item is “I have trouble estimating how long it will take to complete a task”. Materials organization (3 items) is the capacity of an individual to create and maintain systems to stay informed of information or materials, an example of the item is “I lose things”. Emotional regulation (3 items) includes an individual’s ability to manage their emotions so that they might complete tasks, achieve goals, and control or direct their behavior. A sample item is “I get upset when things don’t go as planned”. Lastly, behavior regulation (4 items) is one’s potential to exhibit self-control and considering the consequences of their actions before responding, a sample item is “I say things without thinking”. The average score for each of the five factors can be used to pinpoint areas of executive deficiency, while the total score (summation of the total scores of the five factors) can be used to determine general executive dysfunction. A higher score indicates higher level of executive functioning deficits. 

In the Malaysian context, some of the items (e.g., “I have a short fuse”) are not commonly used, hence we adapted the items (here forth, alternative statements) to ease understanding (see Table 1 for details). A pretest study was conducted to understand if the alternative statements are preferable. A total of 19 working adults who fulfilled the inclusion criteria were invited (using convenience sampling) to read both the original items and alternative statements and choose the one that is easier to understand. Ten (seven Malaysians and three foreigners who were working in Malaysia from different sectors such as academia, healthcare, and management) accepted our invitation to review the alternative statements. An alternative statement is preferable if more than five reviewers indicate that it is superior to the original item. Based on the results, the alternative statements for items 4, 6, 19, and 24 were desirable. Therefore, the alternative statement of these four items was used with the other original items to collect data in the actual study.

Executive Function Index (EFI) [39]. EFI is a self-rating scale to measure EFs within a healthy population. It consists of 27 items measured on a 5-point Likert scale (1 = Not at all, 5 = Very much) to capture five factors: motivational drive, organization, strategic planning, impulse control, and empathy. Thirteen items were reverse scored before computing the total score of the 27 items scores. A higher score indicates better executive functioning.

Self-Rated Creativity Scale (SRCS). Tan and Ong [44] adapted Zhou and George’s [45] supervisory-ratings creativity scale for individuals to self-report their creativity. Individuals responded to the 13 items (e.g., “*I am a good source of creative ideas*”) using a 5-point Likert scale (1 = Strongly Disagree, 5 = Strongly Agree). A higher average score indicates greater creativity [44]. 

9-item Utretch Work Engagement Scale (UWES-9). Schaufeli and Bakker [46] developed the short version of the Utretch Work Engagement Scale (UWES) [46] to assess employees’ engagement in work. The 9 items were scored on a 7-point Likert scale (0 = Never, 6 = Always/Everyday). The items are grouped into three subscales: vigour (e.g., “At my work, I feel bursting with energy”), dedication (e.g., “I am enthusiastic about my job”), and absorption (e.g., “I am immersed in my work”). A higher (mean) score indicates greater engagement.

Employee Well-being Scale (EWB) [47]. The EWB consists of 18 items and was scored on a 7-point Likert scale (1 = Strongly Disagree, 7 = Strongly Agree). The EWB has three subscales: life well-being (LWB; e.g., “*My life is very fun*”), workplace well-being (WBW; e.g., “*Work is a meaningful experience for me*”); and psychological well-being (PWB; e.g., “*I handle daily affairs well*”). The average score of the items is used to gain insights on the individual’s well-being, a higher score indicating better well-being.

### 2.3. Analytical Plan

The data were analyzed using JASP (ver. 0.11.1, University of Amsterdam, Amsterdam, The Netherlands) and IBM SPSS (ver. 22.0, IBM, New York, NY, USA). Both descriptive statistics (e.g., mean, standard deviation, and percentage of the demographics) and inferential statistics were assessed. Confirmatory Factor Analysis (CFA) was conducted using the Diagonally Weighted Least Squares (DWLS) estimator to examine if the suggested 5-factor structure retains in the Malaysian context. The DWLS estimator was preferred because it outperforms the maximum likelihood (ML) estimator, by yielding more accurate model inferences, and when data are to some extent asymmetric, it is more likely to detect structural relationships [48]. Additionally, unlike ML, DWLS does not require a large sample size [48]. The criteria used to assess the fit of the model were: evaluation of the overall chi-square to degrees of freedom ratio (less than three), Comparative Fit Index (CFI; more than 0.95), Tucker-Lewis Index (TLI; more than 0.95); root-mean-square-error of approximation (RMSEA; less than or equal to 0.05), and the standardized root mean square residual (SRMR; less than 0.08) [49,50].

Cronbach’s alpha and McDonald’s Omega were used to assess the reliability of the ESQ-R and other measurements used in the present study. The results were reported in the Results section. Pearson correlation was conducted to examine the construct validity (i.e., convergent and discriminant validity) and criterion-related validity (i.e., concurrent and incremental validity) of the ESQ-R. Specifically, the EFI [39] was used to test for convergent validity, while discriminant validity was tested by examining the relationship between the ESQ-R and the SRCS scores because the literature suggests that different components of EF are related to creativity (e.g., [27,33,51]). Therefore, it is essential to ensure that the ESQ-R is measuring EF but not creativity. 

The relationship between the ESQ-R and UWES-9 scores was tested to evaluate the concurrent validity (of the ESQ-R) because studies have found associations between EF and work engagement (e.g., [14,16]). Finally, hierarchical multiple regression was employed to test the incremental validity of the ESQ-R. Employee well-being was selected as the outcome variable because studies have found that EFs and the respective components such as emotional regulation are conducive to well-being (e.g., [3,4,52,53]). Demographic variables (e.g., age, gender, years working) were entered in step 1, while creativity and work engagement scores were entered in step 2. The ESQ-R score was entered in step 3 to examine if EFs can contribute to well-being after controlling for the effect of demographics, creativity, and work engagement. The incremental validity (of the ESQ-R) is supported if the ESQ-R score shows a statistically significant relationship with the well-being score. 

## 3. Results

### 3.1. Confirmatory Factor Analysis 

Several CFAs using the DWLS estimator were conducted to compare three competing models (e.g., single-factor model, 5-factor model, and 5-factor second-order model) and determine the model of best fit. All models showed a good fit (see Table 2). Theory insinuates the development of a second-order model for EFs (e.g., [1,54,55]). Moreover, as demonstrated in Table 3, the five factors are strongly correlated with each other, consequently supporting the higher-order factor structure. Hence, the 5-factor second-order model (see Figure 1) is preferable. The (standardized) factor loadings were statistically significant and were greater than 0.47 except for two items of the behavioral regulation subscale.

### 3.2. Reliability and Validity

Table 3 shows the descriptive statistics, correlation between variables, and reliability of the ESQ-R and measures used in the present study. The ESQ-R showed excellent internal consistency (α = 0.901; ω = 0.907), while the five factors of the ESQ-R except for the behavioral regulation factor showed adequate reliability. Similarly, the other measures also demonstrated good internal consistency.

Pearson correlation analyses were conducted to examine the construct and criterion validity of the ESQ-R. Results showed a negative correlation between the ESQ-R and EFI scores. As the ESQ-R measures EFs deficiency while the EFI scores indicate EFs strength, the negative relationship exhibits convergent evidence of the ESQ-R. Similarly, there was a negative correlation between ESQ-R and self-rated creativity. The result is consistent with literature that EF is related to, but distinct from, creativity, supporting the discriminant validity of the ESQ-R. 

The ESQ-R also demonstrated good criterion validity. The concurrent validity was evident by the negative relationship between ESQ-R and UWES-9 scores. Meanwhile, the results of hierarchical multiple regression analysis (see Table 4) support the incremental validity. In particular, Model 3 was significant, *F*(6, 287) = 42.847, *p* < 0.001, and explained for 47.30% of well-being. The ESQ-R score continues to have a negative relationship with well-being even after excluding the effect of work engagement and creativity.

## 4. Discussion

This research investigated the psychometric qualities of the Executive Skills Questionnaire-Revised (ESQ-R) among working adults in Malaysia. Most of the previous EF measurement scales were developed to be used in the clinical settings; hence, the self-rated scale ESQ-R is an ideal choice for the non-clinical context. The findings of the study provide support for the psychometric properties of the scale, hence indicating that the ESQ-R is an appropriate measure of executive functioning in the Malaysian working context.

Confirmatory factor analyses were conducted to determine if the 5-factor structure would be retained in the context of the study. While Strait and colleagues [18] suggest a 5-factor model, our results showed that the 5-factor second-order model with five first-order factors (i.e., plan management, time management, materials organization, emotional regulation, and behavioral regulation) and a general (second-order) factor of EF is most preferable. The advantages of using a second-order model are that a first-order analysis provides a narrow, close-up but detailed view of the data, while the second-order analysis provides a wide and general view, providing a different perspective of the data [56,57,58]. Executive functioning has been commonly described as an umbrella term for various cognitive processes/functions that are responsible for goal-orientation, including self-control, behavioral regulation, planning, and organization skills (e.g., [1,55]). Interestingly, our CFA results also show that the emotional regulation dimension has a relatively lower factor loading than the other dimensions. This finding is consistent with the previous study that employed other executive functions measures [59]. Noteworthy, recent studies had suggested activation of the dorsolateral prefrontal cortex (dlPFC), which linked to executive functions, which somewhat improved regulation of emotional interference in selective attention tasks [60,61,62].

The ESQ-R was found to have excellent reliability in the present study. The internal consistency estimates of all the overall ESQ-R and its subscales were sufficient except for the behavioral regulation subscale. The result is consistent with the study conducted by Strait and colleagues [18] whereby the factor loadings for items 19 and 25 were 0.49 and 0.30, respectively. Therefore, future researchers are suggested to review and further develop the items of the behavioral regulation subscale by either rewording the items to provide a more comprehensive representation of behavioral regulation or by deleting the item. 

The construct validity results of the ESQ-R are also promising. Convergent validity was established by depicting a negative correlation between the ESQ-R and the EFI. Indeed, the relationship (between ESQ-R and EFI scores) is stronger than the relationship of the ESQ-R score with other variables. Note that the score of the ESQ-R indicates individuals’ executive dysfunction while the EFI score shows an individuals’ executive functioning. The negative correlation, therefore, indicates that people who reported a high level of EFs (i.e., a higher score in EFI) tend to report less dysfunction (i.e., lower score) in ESQ-R. Discriminant validity was supported by the low correlation between the ESQ-R score and self-rated creativity score. Consistent with past studies that EFs are an antecedent factor of creativity [33,63], our results indicate that EFs and creativity are two distinct constructs and ESQ-R is not measuring creativity. 

Similarly, the ESQ-R has demonstrated criterion validity. The concurrent validity of the ESQ-R is evident by a negative correlation between the ESQ-R and the UWES-9 scores. In other words, people who reported a lower score in ESQ-R (i.e., less dysfunction) are more likely to engage in work. The result is consistent with past studies that EFs are beneficial to engagement [14,16,64]. Our results also support the incremental validity of the ESQ-R. In line with the literature (e.g., [4,53]), ESQ-R was found to have a significant relationship with well-being above and beyond work engagement and creativity. The negative relationship implies that employees who reported a lower score in ESQ-R (i.e., less dysfunction) tend to have a higher level of well-being. The beneficial effect could be due to EFs being essential to problem-solving skills [5], which has been found to reduce stress and increase well-being [44]. 

To our best knowledge, this is the first study validating the ESQ-R in the Malaysian context. Moreover, unlike past studies that focused on students, the present study examined the suitability of the ESQ-R among working adults. The sound psychometric properties of the scale, namely the strong reliability and validity, attest to the quality of the scale and lend support to its generalizability. As a result, organizations can use the ESQ-R to explore executive functioning within the organization and can depend upon its use for various purposes (e.g., selection and recruitment, promotion, training). Furthermore, the ESQ-R is an intervention-focused measurement tool. Organizations may formulate intervention plans based on the employees’ scores in the ESQ-R, as well as evaluate and monitor the intervention progress. For instance, the ESQ-R can be used in future studies to investigate the relationship of EFs with work engagement and employee well-being in work settings.

## 5. Limitations and Suggestions

Limitations of the study are that the data were mostly collected from the education and related industries (70.9%; e.g., lecturers, teachers, tutors, etc.), confining the sample to a certain group with similar demographics, hence restricting the generalizability of the results. It is recommended that future studies focus on greater representativeness by replicating the study with participants from various industries. Additionally, it is beneficial to assess the cross-cultural applicability of the ESQ-R by recruiting participants from across various ethnic groups.

It is noteworthy that the predictive validity of the ESQ-R was not examined. It is critical to ensure the predictability of the ESQ-R [65] for organizations to apply the ESQ-R in recruitment and selection.

## 6. Conclusions

The present study sheds light on the psychometric qualities and usefulness of ESQ-R. The (modified) ESQ-R is best represented by a second-order model with five specifics (first-order) factors and a general (second-order) factor of EFs and demonstrates good internal consistency and (construct and criterion) validity. The ESQ-R is a helpful quantitative tool for measuring executive functioning of employees. Moreover, organizations may utilize the ESQ-R in hiring and promotion. Specifically, the results of ESQ-R can serve as a reference for organizations to identify individuals who have the required executive function abilities for the particular position. In the same vein, the scale may be used by organizations to formulate intervention programs for employees with deficits in executive functioning and evaluate effectiveness of the intervention.

## Figures and Tables

**Figure 1 ijerph-18-08978-f001:**
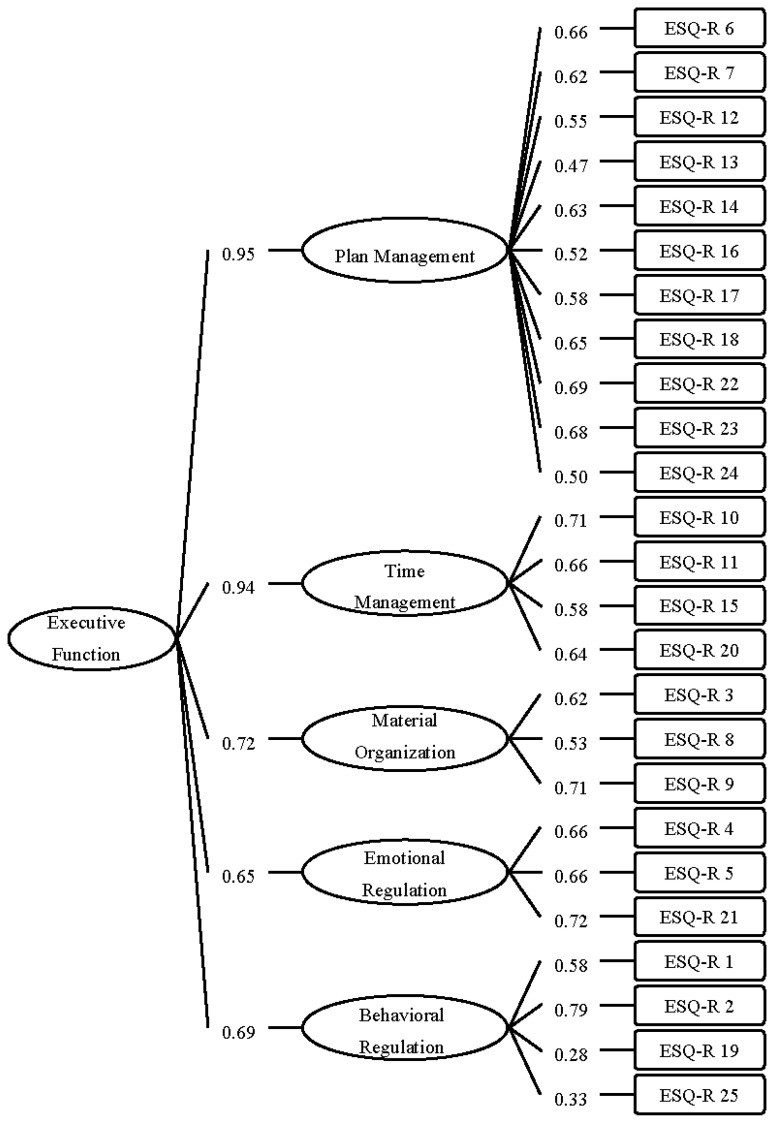
The 5-factor Second-Order Model of the Executive Skills Questionnaire-Revised.

**Table 1 ijerph-18-08978-t001:** Alternative Statements for the Selected Executive Skills Questionnaire-Revised.

Item Number	Original Statement	Alternative Statement
1	I act on impulse.	I act on the spot without planning it.
4 *	I have a short fuse.	I tend to get angry easily.
6 *	I run out of steam before finishing a task.	I lose energy or interest before completing a task.
19 *	I “go with my gut” when making decisions.	I trust my instincts when making decisions.
20	I get so wrapped up in what I’m doing that I forget about other things I need to do.	I’m so focused on what I’m doing that I forget about other things I need to do.
22	I have trouble getting back on track if I’m interrupted.	I have trouble continuing work as planned if I’m interrupted.
24 *	I miss the big picture.	I overlook the whole scenario.
25	I live in the moment.	I concentrate only on the present situation.

* The alternative statement was used in the actual study.

**Table 2 ijerph-18-08978-t002:** Goodness-of-fit indices for Executive Skills Questionnaire-Revised.

Model	*χ* ^2^	df	*χ*^2^/df	CFI	TLI	RMSEA [90% CI]	SRMR
1. 1-factor model	385.170	275	1.401	0.982	0.980	0.035 [0.026, 0.043]	0.071
2. 5-factor model	247.627	265	0.934	1.000	1.003	0.000 [0.000, 0.016]	0.056
3. 5-factor second order model	277.782	270	1.029	0.999	0.999	0.009 [0.000, 0.024]	0.059

Note. N = 325. CFI = comparative fit index; TLI = Tucker-Lewis index; RMSEA = root-mean-square error of approximation; SRMR = standardized root mean square residual; CI: confidence interval. Analysis ran using Diagonally Weighted Least Squares (DWLS) estimator.

**Table 3 ijerph-18-08978-t003:** Descriptive and Intercorrelations of Study Variables.

Variable	N	M	SD	1	2	3	4	5	6	7	8	9	10
1. ESQ-R	325	20.54	10.268	1									
2. Plan Mgt	325	7.59	5.077	0.923 ***	1								
3. Time Mgt	325	3.03	2.241	0.830 ***	0.728 ***	1							
4. Material	325	2.57	1.958	0.640 ***	0.468 ***	0.555 ***	1						
5. Emotional	325	3.02	1.876	0.646 ***	0.497 ***	0.380 ***	0.322 ***	1					
6. Behavioral	325	4.32	2.081	0.605 ***	0.440 ***	0.377 ***	0.189 ***	0.364 ***	1				
7. EFI	308	101.75	11.197	−0.633 ***	−0.630 ***	−0.502 ***	−0.392 ***	−0.394 ***	−0.339 ***	1			
8. Creativity	303	3.79	0.620	−0.329 ***	−0.422 ***	−0.238 ***	−0.133 *	−0.197 ***	−0.048	0.437 ***	1		
9. Engage	305	4.26	1.040	−0.365 ***	−0.365 ***	−0.327 ***	−0.198 ***	−0.260 ***	−0.134 *	−0.520 ***	0.431 ***	1	
10. EWB	300	5.41	0.995	−0.447 ***	−0.438 ***	−0.38 6***	−0.291 ***	−0.346 ***	−0.147 *	0.517 ***	0.441 ***	0.612 ***	1
⍺				0.901	0.860	0.745	0.693	0.732	0.567	0.802	0.939	0.930	0.952
ω				0.907	0.862	0.748	0.726	0.741	0.597	0.813	0.941	0.934	0.955

Note. M = mean; SD = standard deviation; ESQ-R = overall score of the Executive Skills Questionnaire-Revised (a higher score indicates a lower executive functioning); Plan Mgt = Plan Management; Time Mgt = Time Management; Material = Material Organization; Emotional = Emotional Regulation; Behavioral = Behavioral Regulation; EFI = Executive Function Index; Creativity = Self-Rated Creativity; Engage = Work Engagement; EWB = Employee Well-Being; ⍺ = Cronbach alpha coefficient; ω = McDonald omega coefficient. Missing values were handled using exclude case pairwise (N = 299 to 325). ** p* < 0.05, *** *p <* 0.001.

**Table 4 ijerph-18-08978-t004:** Results of Hierarchical Multiple Regression Analysis for Executive Functions.

	B	SE	β	*p*	95% CI [LLCI, ULCI]	VIF	ΔR^2^	ΔF
Step 1							0.037	3.712 *
Constant	4.611	0.249		>0.001	[4.120, 5.102]			
Age	0.021	0.007	0.222	0.004	[0.007, 0.036]	1.801		
Gender	0.048	0.117	0.024	0.682	[−0.183, 0.279]	1.071		
Years	−0.009	0.010	−0.070	0.357	[−0.028, 0.010]	1.718		
Step 2							0.390	97.847 ***
Constant	1.997	0.320		>0.001	[1.368, 2.626]			
Age	0.004	0.006	0.047	0.445	[−0.007, 0.016]	1.880		
Gender	0.014	0.092	0.007	0.880	[−0.168, 0.196]	1.105		
Years	−0.002	0.007	−0.020	0.737	[−0.017, 0.012]	1.725		
Engagement	0.528	0.048	0.555	>0.001	[0.433, 0.623]	1.289		
Creativity	0.263	0.082	0.162	0.002	[0.101, 0.424]	1.285		
Step 3							0.046	24.977 ***
Constant	3.022	0.369		>0.001	[2.296, 3.479]			
Age	0.002	0.006	0.025	0.676	[−0.009, 0.013]	1.891		
Gender	−0.005	0.089	−0.003	0.954	[−0.180, 0.170]	1.107		
Years	−0.002	0.007	−0.019	0.734	[−0.016, 0.012]	1.725		
Engagement	0.473	0.048	0.497	>0.001	[0.380, 0.567]	1.362		
Creativity	0.201	0.080	0.124	0.012	[0.044, 0.358]	1.317		
ESQ-R	−0.023	0.005	−0.233	>0.001	[−0.032, −0.014]	1.183		

Note. N = 294. SE = standard error; CI = confidence interval; LLCI = lower limit confidence interval; ULCI = upper limit confidence interval; VIF = variance inflation factor; Gender: female as the reference group; Years: years working; ESQ-R = Executive Skills Questionnaire-Revised. ** p* < 0.05, *** *p <* 0.001.

## Data Availability

The datasets generated during and/or analysis during the present study are available from the corresponding author upon request.

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
