# Peer review of "The Executive Skills Questionnaire-Revised: Adaptation and Psychometric Properties in the Working Context of Malaysia"

_ijerph, 2021, doi:10.3390/ijerph18178978_

Round 1

Reviewer 1 Report

The topic of this manuscript was described clearly related to the selection of questionnaires, recruiting of individuals to complete the questionnaire, and the use of both first and second order questions.  I found no spelling or otherwise unclear statements in this document.  The authors' focus on using these evaluation tools in a setting with adult workers seemed innovative since the tests had been used for students in other studies. I recommend that this article be published to that employers might choose this process for identifying candidates for hiring as well as promotion in a specific setting.

Author Response

We thank the reviewer for contributing time to review our manuscript.

Thanks for the encouragement.

Reviewer 2 Report

Overall, the Ms is well written, albeit it requires some minor corrections of the English, and the study is sound. However, I have a few minor comments that will improve the readability and impact of the Ms.

  1. Lines 57-58: the sentence “In the workplace, EFs refer to the interaction between intellect, perceptions, and reasoning for performing daily functions (Bade, 2010)” gives the false impression of an equivalence between executive functions and IQ (in lines 152-153 is clarified that there is a general misunderstanding that executive functions and general intelligence are interchangeable). Please, rephrase it to prevent contributing to the misunderstanding.
  2. There are some formatting issues with Figure 1
  3. Line 187: there must be an error with the age-range reported here as the upper age of 80 years old is reported for executive professionals still working (i.e, not retired).
  4. The standardized factor loadings for emotion regulation items (see Figure 1) are somewhat lower (i.e., 0.65) than for other items (i.e., plan management, 0.95 and time management, 0.94 and material organization, 0.72) and similarly lower to the behavioural regulation items (i.e., 0.69). This aspect should be discussed with reference to evidence showing that potentiating the activity of the Dorsolateral Prefrontal Cortex involved in executive functions does improve inhibition of emotional distractors in a selective attention task (e.g., Pecchinenda et al. 2015, Neuropsychologia; Petrucci & Pecchinenda, 2017, Cognition & Emotion; De Luca et al., 2020, Symmetry).

Reviewer 3 Report

First of all, congratulate the authors for the work done, however, there are certain aspects that need to be improved. So, I list them below:

  1. In-text citations must follow the journal's regulations, currently they do not comply. I advise you to review this aspect.
  2. The instruments do not show the coefficient to measure the reliability of the scales used (Cronbach's Alpha), which is why I ask you to include it for each of the instruments used.
  3. Regarding the conclusions section, it is suggested that they include the possible practical implications, providing an explanation of the importance and relevance of the study.
  4. Regarding the references section, you should review and adapt the references to the regulations of the journal, since they do not comply with the regulations.
